# Developing Models to Predict *BRAF*V600E and *RAS* Mutational Status in Papillary Thyroid Carcinoma Using Clinicopathological Features and pERK1/2 Immunohistochemistry Expression

**DOI:** 10.3390/biomedicines11102803

**Published:** 2023-10-16

**Authors:** Agnes Stephanie Harahap, Imam Subekti, Sonar Soni Panigoro, Retno Asti Werdhani, Hasrayati Agustina, Dina Khoirunnisa, Mutiah Mutmainnah, Fajar Lamhot Gultom, Abdillah Hasbi Assadyk, Maria Francisca Ham

**Affiliations:** 1Department of Anatomical Pathology, Faculty of Medicine, Universitas Indonesia, Dr. Cipto Mangunkusumo Hospital, Jakarta 10430, Indonesia; agnes.stephanie01@ui.ac.id (A.S.H.); lisnawatidr@gmail.com (L.); dinakhoirunn@gmail.com (D.K.); mutiaaa20@gmail.com (M.M.); 2Human Cancer Research Center-Indonesian Medical Education and Research Institute, Faculty of Medicine, Universitas Indonesia, Jakarta 10430, Indonesia; 3Doctoral Program in Medical Sciences, Faculty of Medicine, Universitas Indonesia, Jakarta 10430, Indonesia; 4Department of Internal Medicine, Faculty of Medicine, Universitas Indonesia, Dr. Cipto Mangunkusumo Hospital, Jakarta 10430, Indonesia; isubekti@yahoo.com; 5Department of Surgery, Faculty of Medicine, Universitas Indonesia, Dr. Cipto Mangunkusumo Hospital, Jakarta 10430, Indonesia; sonarpanigoro@gmail.com; 6Department of Medical Biology, Faculty of Medicine, Universitas Indonesia, Dr. Cipto Mangunkusumo Hospital, Jakarta 10430, Indonesia; asmarinah.si@gmail.com; 7Department of Community Medicine, Faculty of Medicine, Universitas Indonesia, Jakarta 10310, Indonesia; retnoasti@gmail.com; 8Department of Anatomical Pathology, Faculty of Medicine, Universitas Padjadjaran, Hasan Sadikin General Hospital, Bandung 40161, Indonesia; hasrayati@gmail.com; 9Department of Anatomical Pathology, MRCCC Siloam Hospital, Jakarta 12930, Indonesia; fajar_lamhot@yahoo.com; 10Department of Anatomical Pathology, Faculty of Medicine, Universitas Kristen Indonesia, Jakarta 13630, Indonesia; 11Department of Otorhinolaryngology, Head and Neck Surgery, Harapan Kita National Women and Children Health Center, Jakarta 11420, Indonesia; abdillahhasbiassadyk@gmail.com

**Keywords:** papillary thyroid carcinoma, BRAF-like, RAS-like, *BRAF*V600E, *RAS* mutation, prediction model

## Abstract

The Cancer Genome Atlas (TCGA) has classified papillary thyroid carcinoma (PTC) into indolent RAS-like and aggressive BRAF-like based on its distinct driver gene mutations. This retrospective study aimed to assess clinicopathology and pERK1/2 expression variations between BRAF-like and RAS-like PTCs and establish predictive models for *BRAF*V600E and *RAS*-mutated PTCs. A total of 222 PTCs underwent immunohistochemistry staining to assess pERK1/2 expression and Sanger sequencing to analyze the *BRAF* and *RAS* genes. Multivariate logistic regression was employed to develop prediction models. Independent predictors of the *BRAF*V600E mutation include a nuclear score of 3, the absence of capsules, an aggressive histology subtype, and pERK1/2 levels exceeding 10% (X^2^ = 0.128, *p* > 0.05, AUC = 0.734, *p* < 0.001). The *RAS* mutation predictive model includes follicular histology subtype and pERK1/2 expression > 10% (X^2^ = 0.174, *p* > 0.05, AUC = 0.8, *p* < 0.001). We propose using the prediction model concurrently with four potential combination group outcomes. PTC cases included in a combination of the low-*BRAF*V600E-scoring group and high-*RAS*-scoring group are categorized as RAS-like (adjOR = 4.857, *p* = 0.01, 95% CI = 1.470–16.049). PTCs included in a combination of the high-*BRAF*V600E-scoring group and low-*RAS*-scoring group are categorized as BRAF-like PTCs (adjOR = 3.091, *p* = 0.001, 95% CI = 1.594–5.995). The different prediction models indicate variations in biological behavior between BRAF-like and RAS-like PTCs.

## 1. Introduction

Thyroid carcinomas are among the most prevalent malignancies of the endocrine system, with a notable rise in incidence over the last few decades [1]. The emergence of well-differentiated thyroid tumors such as papillary thyroid carcinoma (PTC) and follicular thyroid carcinoma (FTC) has been linked to alterations in various genes, including *BRAF*, *RAS*, and *RET*, and recently discovered gene fusions, such as *EIFIAX*, *RET*, *NTRK1/3*, *ALK*, *PAX8-PPARG*, *RGADA*, *FGR2*, and *LTK* [2]. PTC, which constitutes 80–85% of overall thyroid carcinoma cases, has particularly received an advanced exploration of its genomic landscape [3]. *BRAF*V600E and *RAS* mutations are the two most prevalent gene mutations detected in PTC, with rates of 28–83% [4,5] and 11.5–20% [3,6,7], respectively. Being mutually exclusive, these driver gene mutations ultimately lead to the same incongruous activation of the mitogen-activated protein kinase (MAPK) pathway [8]. This pathway involves the sequential activation and phosphorylation of RAS, RAF, MEK, and ERK, which are important in the regulation of cellular growth and apoptosis (Figure 1). Dysregulation of the gene associated with this pathway can affect cellular function and promote tumorigenesis. Elevated pERK1/2 expression, detected through immunohistochemistry staining or Western blot analysis, has been considered a proxy indicator for heightened MAPK pathway activity in various malignancies [9]. Interestingly, the expression of pERK1/2 has reportedly differed between *BRAF*-mutated and *RAS*-mutated tumors, with the former being more elevated [2]. In addition, *RAS*-mutated PTC demonstrates concurrent activation of P13K/AKT signaling as well as MAPK activation [2]. These signaling differences result in distinct phenotypes of PTCs, which are characterized by varied clinical and histopathological findings.

Among several *BRAF* mutations that have been identified in PTC, *BRAF*V600E constitutes the most cases. This T1799A point mutation results in the replacement of valine with glutamate and has emerged as a significant clinical determinant due to its association with heightened disease aggressiveness [10]. PTC tumors typically display an inert behavior, with a considerable proportion of patients attaining a survival rate of ten years [11]. However, tumors harboring *BRAF*V600E were associated with increased mortality rates, higher rates of recurrence, and resistance to radioiodine treatment [12,13]. Several studies have also linked this mutation to various aggressive pathology features, such as perithyroidal extension, node metastases, and advanced clinical stage [14,15,16]. In contrast to the *BRAF*V600E mutation, tumors possessing the *RAS* mutation were associated with less aggressive pathology characteristics, involving a follicular histology subtype [12], encapsulated tumors [17], minimal disease invasion [15,18], and lower risk of recurrence [12]. Among the three isoforms of the *RAS* gene, *NRAS* is the most prevalent gene mutation and is closely related to PTC [19]. While it is less common in North American and European populations, the *RAS* mutation is reported more frequently in the Asian population [20].

A recent discovery by The Cancer Genome Atlas (TCGA) has been able to classify PTC based on its two major driver gene mutations, which are BRAF-like and RAS-like tumors [2]. Identifying PTCs into BRAF-like and RAS-like tumors during diagnostic workup is essential, not only for determining the most precise and targeted treatment but also to comprehend the biological behavior between the two. Hence, this study aimed to explore the differences between BRAF-like and RAS-like tumors concerning clinicopathology and pERK1/2 expression and further establish predictive models for *BRAF*V600E and *RAS* mutations in PTC.

## 2. Materials and Methods

### 2.1. Ethical Approval

The study was conducted in accordance with the Helsinki Declaration and was authorized by the Institutional Research Ethics Committee of the Faculty of Medicine, Universitas Indonesia, Dr. Cipto Mangunkusumo Hospital (FMUI-220 CMH) with approval KET-253/UN2.F1/ETIK/PPM.00.02/2022. An ethical waiver of informed consent from the Institutional Review Board was received (permission ND-532/UN2.FI/ETIK/PPM.00.02.2022).

### 2.2. Study Design and Population

This study retrospectively enrolled PTC patients who had undergone total thyroidectomy at Dr. Cipto Mangunkusumo National Hospital and MRCCC Siloam Hospital between January 2019 and September 2022. We excluded cases with high-grade features, such as a high mitotic index (>3 per 2 mm^2^) and/or necrosis. The clinical information, including age, gender, and clinical stage, was procured from medical records. Three licensed pathologists blindly gathered the histopathological data: tumor size, PTC nuclear score (Appendix A), capsule, histology subtype, multifocality, lymphovascular invasion (LVI), extrathyroidal extension (ETE), and node metastases. Histology subtypes of PTCs were further classified into non-aggressive (classic and follicular) and aggressive (tall cell, oncocytic, and solid) [13]. The protocol of the present study is illustrated in Figure 2.

### 2.3. pERK1/2 Immunohistochemistry Examination

The expression of pERK1/2 in this present study was classified into high and low expression based on the cutoff point of 10% established by Gomes et al. [21]. The standard immunohistochemical evaluation procedures were used to assess the expression of pERK1/2. A positive and negative control was included in each specimen. Colon adenocarcinoma paraffin blocks as a positive control were taken from the routine control archives in our institution. The negative controls are tissue samples that did not receive any application of primary antibody reagents. Unstained slides 3 mm thick were cut and rinsed under running water for 2 min following deparaffinization and rehydration. In a de-cloaking chamber at a temperature of 96 °C for 25 min, antigen retrieval was carried out using pH 9 Tris-EDTA buffer. After 3 min of washing in PBS pH 7.4, a blocking solution (Leica Cat. No: RE7102-CE, Thermo Fisher Scientific, Inc., Waltham, MA, USA) was administered for 20 min at room temperature to block non-specific protein. Each slide was incubated with rabbit monoclonal anti-Phospho-p44/42 MAPK (Erk1/2) (20G11; Cell Signaling Technology, Danvers, MA, USA) at a dilution ratio of 1:600. Subsequently, each slide was incubated with the PolyVue Plus Mouse/Rabbit Enhancer (Diagnostic Biosystems, Pleasanton, CA, USA) for 15 min followed by PolyVue Plus Mouse/Rabbit HRP Label for 15 min. The slides were repeatedly washed before being incubated to diluted diaminobenzidine chromogen buffer substrate for 1 min at room temperature. Mayer’s hematoxylin was used for a 10 s counterstaining procedure at room temperature. Each slide was examined under a light microscope (Leica Microsystems GmbH, Wetzlar, Germany) and photographed in five representative fields at ×400 magnification with a minimum of 500 tumor cells for each case. Tumor cells stained brown in the nucleus were counted as positive. The quantitative evaluation of pERK1/2 expression was performed by counting the proportion of cells stained positively using ImageJ software version 1.51 (National Institutes of Health). Kappa interobserver analysis indicated an agreement of 0.879 (*p* < 0.001), which is near perfect.

### 2.4. Mutational Analysis

The tumor specimens were subjected to mutational analysis using Sanger sequencing to detect the *BRAF*V600E mutation as well as *N/H/K-RAS* codon 12, 13, and 61 mutations [2]. The procedures were performed according to the methodology outlined in a previous study [22]. Non-BRAF/non-RAS mutations are cases with neither *BRAF*V600E nor *RAS* mutation, including *N/H/K-RAS* mutations observed in the respective gene and codon.

### 2.5. Statistical Analysis

The statistical software package SPSS version 20 was utilized for the purpose of data processing. Bivariate analyses were conducted utilizing Chi-squared and Mann–Whitney U tests. Clinicopathological variables that showed a *p* value < 0.05 during bivariate analysis were considered as significant variables and were subsequently added to a multivariate analysis. Binary logistic regression testing was employed to conduct the multivariate analysis utilizing a backward conditional method. The model’s goodness of fit was evaluated by conducting the Hosmer–Lemeshow test. A significance level of 0.05 or higher is indicative of a reliable predictive model. The fittest model resulted in selected clinicopathological variables that act as predictors for each *BRAF*V600E and *RAS* mutational status. The development of a scoring value for each predictor involved the formulation of the coefficient B and S.E. as displayed in the regression test. Each predictor variable displayed a different B coefficient and S.E. The first step of developing the scoring system would be dividing the B coefficient by S.E. (B/S.E. value) of each variable. After attaining the B/S.E. value for each predictor variable, the lowest B/S.E. value was determined. The final score of each variable was then obtained by dividing each respective B/S.E. value to the lowest B/S.E. value. Following the implementation of the scoring system for the study sample, an analysis of the receiver-operating characteristic (ROC) curve was conducted. An area under the receiver-operating characteristic curve (AUC) exceeding 0.7 indicates a satisfactory level of diagnostic precision. The scoring wizard tool was utilized to evaluate the probability of each total score in every predictive model. To evaluate the applicable combinations of *BRAF*V600E and *RAS* model prediction, a multinomial logistic regression test and internal validation was performed.

## 3. Results

### 3.1. Baseline Characteristics

The study retrospectively collected PTC patients who had undergone total thyroidectomy from January 2019 to September 2022, with an initial recruitment of 527 patients. A total of 305 patients were excluded from the study for multiple reasons, as illustrated in Figure 3. The study consisted of a total of 222 participants.

Demographic information gathered included female sex (162, 73%), diagnosed under the age of 55 years (163, 73.4%), and at clinical stage 1 (166, 74.8%). Histopathological features were dominated by cases characterized by tumor size less than 4 cm (166, 74.8%), nuclear score of 3 (152, 68.5%), lack of capsules (139, 62.6%), and non-aggressive subtypes (152, 68.5%). The histology subtypes were dominated by follicular subtype (87, 39.2%), followed by classic (65, 29.3%), tall cells (53, 23.9%), oncocytic (10, 4.4%), and solid (7, 3.2%), respectively. The presence of multifocality (168, 75.7%), absence of LVI (135, 60.8%), lack of ETE (158, 71.2%), and lack of node metastases (134, 60.4%) constituted the majority of cases in this study. The expression of pERK 1/2 was quantified to range from 0.2% to 99%, with a median value of 5%. Based on the 10% cutoff point [21], this study was dominated by low pERK1/2 expression (145, 65.3%). Figure 4 displays histological findings of this study sample.

### 3.2. Bivariate Analysis: Correlation between Clinico-Histopathology Characteristics with BRAFV600E and RAS Mutational Status

One hundred and sixteen cases that did not exhibit the *BRAF*V600E or *RAS* mutation were further designated as controls. Two distinct bivariate analyses were performed to compare the *BRAF*V600E-mutated and controls, as well as the *RAS*-mutated and controls. The results are outlined in Table 1 and Table 2. The bivariate analysis conducted on the *BRAF*V600E-mutated group revealed a significant correlation between *BRAF*V600E mutation and nuclear score 3 (*p* = 0.001; OR = 4.1; 95% CI = 1.7–9.4), the absence of tumor capsules (*p* = 0.001; OR = 3.2; 95% CI = 1.5–6.7), tall cell subtype (*p* = 0.001; OR = 8.9; 95% CI = 3.5–22.6), aggressive histology subtypes (*p* = 0.001; OR = 2.9; 95% CI = 1.5–5.4), the presence of ETE (*p* = 0.01; OR = 2.3; 95% CI = 1.2–4.4), the presence of node metastases (*p* = 0.008; OR = 2.3; 95% CI = 1.2–4.3), and high pERK1/2 expression (*p* = 0.008; OR = 2.4; 95% CI = 1.2–4.8). There was a significant association between *RAS* mutation and the follicular histology subtype (*p* = 0.001; OR = 2.6; 95% CI = 1.1–6.2), the non-aggressive histology subtypes (*p* = 0.001; OR = 17; 95% CI = 2.2–128.6), and high pERK1/2 expression (*p* = 0.001; OR = 7.6; 95% CI = 3.5–16.7).

### 3.3. Multivariate Analysis: Establishing the BRAFV600E Prediction Model

A multivariate analysis using logistic regression was conducted to examine the association between the *BRAF*V600E mutation and multiple variables. A nuclear score of 3, the absence of tumor capsules, aggressive histology subtypes, and high pERK1/2 expression were identified as predictive factors contributing to the presence of the *BRAF*V600E mutation. As indicated in Table 3, the predictor variables were assessed individually to determine their respective score for the development of a *BRAF*V600E prediction model. A nuclear score of 3, the lack of tumor capsules, and aggressive histology subtypes each contribute a score of 1. pERK1/2 expression level exceeding 10% corresponds to a score of 2.

The Hosmer–Lemeshow goodness-of-fit test yielded results indicating that the logistic regression model exhibited a favorable level of calibration (X^2^ = 0.128, *p* > 0.05). The receiver-operating characteristic (ROC) curve’s area under the curve (AUC) was determined to be 0.734 with *p*-value < 0.001 and a 95% CI of 0.661–0.807 (Figure 5). This finding suggests that the logistic regression model exhibits a favorable level of discrimination. Probability, sensitivity, and specificity values for every outcome of the overall score are summarized in Table 4. Based on the results of probability analysis, it was determined that the highest probability, amounting to 82%, is associated with a total score of 5. This indicates that if PTC achieves a score of 5, there is an 82% likelihood of the occurrence of the *BRAF*V600E mutation.

### 3.4. Multivariate Analysis: Establishing the RAS Mutation Prediction Model

Based on the multivariate analysis of *RAS* mutational status, the predictor variables that were included for the development of the *RAS* prediction model were follicular histology subtype and pERK1/2 expression exceeding 10%. Each corresponding variable gives a score of 1 based on the B/SE value, as summarized in Table 5.

The result of the Hosmer–Lemeshow goodness-of-fit test suggests that the RAS mutation logistic regression model demonstrated a satisfactory level of calibration (X^2^ = 0.174, *p* > 0.05). The AUC of the ROC curve was found to be 0.8, with a *p*-value < 0.001 and a 95% CI of 0.702–0.854 (Figure 6). Probability, sensitivity, and specificity values for every outcome of the overall score generated are summarized in Table 6. According to the findings of the probability analysis, it was ascertained that the highest probability, 70%, is linked to a cumulative score of 2. This suggests that in the case of PTC obtaining a score of 2, there is a probability of 70% of the presence of the *RAS* mutation.

### 3.5. Internal Validation: Applying BRAFV600E and RAS Mutation Prediction Model to Study Samples

Both prediction models were used to internally validate all study samples. The results showed that a sample capable of fulfilling two prediction models had varying probabilities. Consequently, we established four possible combination outcomes based on the scores obtained from the model of the combination of *BRAF*V600E and *RAS* mutations (Figure 7).

The *BRAF*V600E prediction model results were classified into a low-*BRAF*V600E-scoring group (total score 0–2) and high-*BRAF*V600E-scoring group (total score 3–5) based on a specificity value of 65% as the middle threshold for identifying *BRAF*V600E mutational status. The *RAS* prediction model results were classified into a low-*RAS*-scoring group (total score of 0–1) and high-*RAS*-scoring group (total score of 2) using a specificity value of 91%.

Table 7 provides a summary of a multinomial analysis on four combination outcomes. The low-*BRAF*V600E-scoring group and low *RAS*-scoring group, combination 1, acted as the reference group, since they had the greatest proportion of non-*BRAF*V600E and non-*RAS* patients. Combination 2 (adjOR = 4.857, *p* = 0.01, 95% CI = 1.470–16.049), low-*BRAF*V600E-scoring group and high-*RAS*-scoring group, was substantially linked to more occurrences of *RAS* mutation and considered a RAS-like combination. A strong correlation existed between the *BRAF*V600E mutation and combination 3, the high-*BRAF*V600E-scoring group and low-*RAS*-scoring group (adjOR = 3.091, *p* = 0.001, 95% CI = 1.594–5.995) and further considered a BRAF-like combination. Combination 4, the high-*BRAF*V600E-scoring group and high-*RAS-*scoring group, was found to have significantly more *RAS*-mutated patients (adjOR = 14.571, *p* = 0.001, 95% CI = 4.095–51.855).

### 3.6. Correlation between Combination Groups with Clinical Endpoint of PTC

Further analysis to assess the correlation between combination groups with the clinical endpoint of PTC were performed. The clinical endpoint assessed in this present study includes clinical stage and node metastasis. As presented in Table 8, there were significant correlations between combination groups with clinical stage (*p =* 0.008) and node metastasis (*p* < 0.001). Combination 2 (RAS-like) was correlated with early clinical stage (adjOR = 1.162, 95% CI = 1.079–1.250), whereas combination 3 (BRAF-like) was correlated with the presence of node metastasis (adjOR = 4.326, 95% CI = 2.330–8.033). 

## 4. Discussion

Various genetic alterations have been identified as contributing factors to the development of PTC. The most commonly observed genetic changes in PTC are *BRAF*V600E and *RAS* mutations [23]. These driver gene mutations are mutually exclusive and contribute to the aberrant activation of the MAPK pathway [24]. The different signaling cascades associated with *BRAF*V600E and *RAS* mutations give rise to the specific phenotypic and behavioral characteristics observed in PTC. Tumors harboring *BRAF*V600E mutations exhibit a greater propensity for aggressiveness, characterized by an increased likelihood of disease recurrence [25], mortality [12], and resistance to radio-ablation [13]. Conversely, tumors carrying *RAS* mutations tend to display more indolent behavior [18]. TCGA has emphasized the importance of categorizing PTCs into two distinct subtypes, namely, BRAF-like and RAS-like, according to their distinctive biological behaviors [2,26]. This present study aimed to develop a predictive model for *BRAF*V600E and *RAS* mutations in PTC using various histopathological features, including the novel PTC nuclear score and pERK1/2 expression.

Histopathological factors that are known to contribute to the disease’s aggressiveness are the presence of tumor multifocality, vascular invasion, perithyroidal soft-tissue invasion, and node metastases [26]. The present study provides more evidence for prior research [27,28] that has established an association between these parameters and *BRAF*V600E mutation status. A phospho-specific antibody that detects pERK1/2 was also examined in this study to assess the activation of the MAPK pathway on a cellular level. It was documented that pERK1/2 expression exceeding 10% was associated with a higher risk of *BRAF*V600E mutation. Jung et al. discovered a correlation between BRAF-like tumors and high nuclear scores [16]. The findings of this current investigation align with those of a prior study, which demonstrated an association between PTC nuclear score of 3 and the *BRAF*V600E mutation. We identified four features that emerged as significant predictors of the presence of the *BRAF*V600E mutation. These variables include a nuclear score of 3, aggressive histology subtypes, the lack of a tumor capsule, and an expression level of pERK1/2 greater than 10%. The scoring system was established utilizing the characteristics indicated in Table 3. A nuclear score of 3, the lack of a tumor capsule, and the aggressive histology subtypes each contributed a score of 1. If the expression of pERK1/2 exceeds 10%, it is assigned a score of 2. The current research demonstrates that the higher the total score, the higher the probability of being *BRAF*V600E-mutated. The probability of the *BRAF*V600E mutation is highest at 82% when a set of total five scores is taken into account.

In comparison to the *BRAF*V600E mutation, tumors harboring the *RAS* mutation have been linked to a less aggressive pathological phenotype, characterized by a follicular-patterned tumors, encapsulated tumors [17], less disease invasion [15,18] and a reduced likelihood of recurrence [18]. Our finding is in line with previous literature, in which *RAS* mutations were significantly more common in the follicular subtype of PTCs. Follicular subtype of PTCs are considered the non-aggressive histology subtypes, belonging to the well-differentiated thyroid carcinomas [13]. A significant difference between the signaling pathways of *BRAF*V600E-mutated and *RAS*-mutated tumors resides in the lower level of MAPK activity found in *RAS*-mutated tumors [2]. This present study, however, was able to display a significant association between the enhanced MAPK activity in both *BRAF*V600E- and *RAS*-mutated PTCs, as displayed in pERK1/2 immunohistochemistry expression. On multivariate analysis, it was determined that two features, namely, the presence of the follicular subtype and an expression level of pERK1/2 greater than 10%, were significant predictors of *RAS* mutation. The scoring system was constructed based on the criteria listed in Table 5. Each variable is assigned a value of 1, resulting in a total score of 2, which contributes to a probability of 70% of *RAS* mutation.

The provided study sample exhibits the capacity to satisfy two distinct prediction models with differing probabilities for *BRAF*V600E and *RAS* mutations, posing challenges in determining mutational status. Hence, this work proposes the concurrent utilization of the *BRAF*V600E and *RAS* prediction models in routine clinical applications. All samples were applied to both the *BRAF*V600E and *RAS* prediction models for internal validation. The initial utilization of the *BRAF*V600E prediction model involved its categorization into two distinct groups: a low-*BRAF*V600E-scoring group (score 0–2) and a high-*BRAF*V600E-scoring group (score 3–5). The *RAS* prediction model consists of two distinct sample groups based on their total scores. One group is characterized by RAS scores ranging from 0 to 1, the low-*RAS*-scoring group, while the other group has a uniform RAS score of 2, the high-*RAS-*scoring group. Ultimately, four possible outcomes were established, denoted as combinations within the context of this investigation (Figure 5). There are four possible outcome groups: combination 1 involves cases with low scores in both the *BRAF*V600E and RAS groups, combination 2 involves cases with a low *BRAF*V600E score and a high *RAS* score, combination 3 involves cases with a high *BRAF*V600E score and a low *RAS* score, and combination 4 involves cases with high scores in both the *BRAF*V600E and *RAS* prediction models. The prevalence of combination 1 was seen to be highest among individuals with non-*BRAF*V600E and non-*RAS* mutations. Combination 2 exhibited the highest prevalence in samples where *RAS* mutations were detected, with a statistically significant positive correlation. In combination 3, most of the samples exhibited *BRAF*V600E mutations, which were shown to be statistically significant. Based on the obtained results, it was determined that the dominant parts of combination 2 and combination 3 were RAS-like and *BRAF*V600E-like PTCs, respectively. This discovery provides evidence in favor of the proposed hypothesis. In combination 4, a significant association was shown between *RAS* mutations and a 14-fold increased likelihood compared to non-*BRAF*V600E non-*RAS* mutations. However, no association was found with *BRAF*V600E mutations. On the one hand, this combination exhibits a proclivity toward *RAS* mutations. However, given that PTC with the *BRAF*V600E mutation tends to display a more aggressive nature, it is advisable to approach its interpretation with caution to avoid potential undertreatment. Undertreatment refers to a situation in which PTC has aggressive characteristics, although the management approach is based on low-risk criteria, thereby elevating the likelihood of disease recurrence or metastasis.

This is the sole study to have constructed a predictive model pertaining to gene mutations in PTC, owing to the routine implementation of molecular examination in developed countries. The outcomes of this study may be beneficial to be implemented in countries with limited access and facilities to molecular testing. Our findings can map the histopathology characteristics of PTC into BRAF-like and RAS-like tumors as a foundation of the biological behavior of the tumor. Although this study was able to present the significant correlations between combination groups with clinical stage and node metastases, it is limited, as it does not include other clinical endpoint variables such as mortality, recurrence, distant metastases, or therapy response. Additional external validation studies are required to further assess the predictive model, utilizing larger and more diverse samples as well as incorporating additional variables, as previously mentioned.

## 5. Conclusions

Using clinico-histopathology features and pERK1/2 expression, two distinct predictive models for *BRAF*V600E and *RAS* mutational status in PTC were developed. The *BRAF*V600E prediction model consists of a PTC nuclear score of 3 (score 1), a lack of capsules (score 1), the aggressive histology subtypes (score 1), and pERK1/2 expression > 10% (score 2). The probability of the *BRAF*V600E mutation is highest at 82% when a set of a total five scores was reached. The *RAS* prediction model consists of the follicular subtype (score 1) and pERK1/2 expression > 10% (score 1). BRAF-like tumors are those included in combination 3 (high-*BRAF*V600E-scoring group and low-*RAS-*scoring group), which exhibits a significant threefold increase in the *BRAF*V600E mutation. RAS-like tumors are those belonging to combination 2 (low-*BRAF*V600E-scoring group and high-*RAS*-scoring group), which showed a significant 4.8-fold increase in *RAS* mutation. Combination 2 (Ras-like) was associated with early clinical stage, whereas combination 3 (BRAF-like) was associated with the presence of node metastasis. These prediction models may serve as a fundamental basis for comprehending the distinct phenotypic and molecular characteristics of BRAF-like and RAS-like PTCs.

## Figures and Tables

**Figure 1 biomedicines-11-02803-f001:**
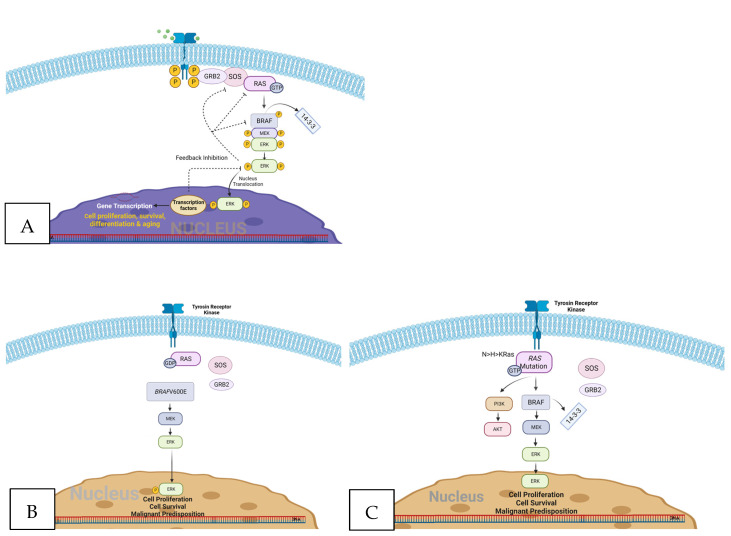
(**A**) The activation of the normal MAPK pathway occurs upon the binding of an extracellular ligand to the receptor tyrosine kinase. This signal triggers the activation of RAS and its downstream effector RAF, leading to the phosphorylation of MEK and ERK. pERK translocates to the nucleus and activates transcription factors, leading to gene transcription. (**B**) Mutated BRAF can independently activate the MAPK pathway without the need for ligand binding or RAS activation. This mutation leads to elevated pERK expression due to reduced sensitivity to feedback inhibition. (**C**) Mutated RAS activates the MAPK pathway independently of ligand binding. RAS additionally triggers the activation of RAF and PI3K/AKT signaling pathways. Created with Biorender.com available at (**A**) https://app.biorender.com/illustrations/640855dca9e401e58f696a60 (**B**) https://app.biorender.com/illustrations/63dbb90b80e0595aad605b56 (**C**) https://app.biorender.com/illustrations/63dd02150cdc41b887bf5865 (accessed on 8 October 2023).

**Figure 2 biomedicines-11-02803-f002:**
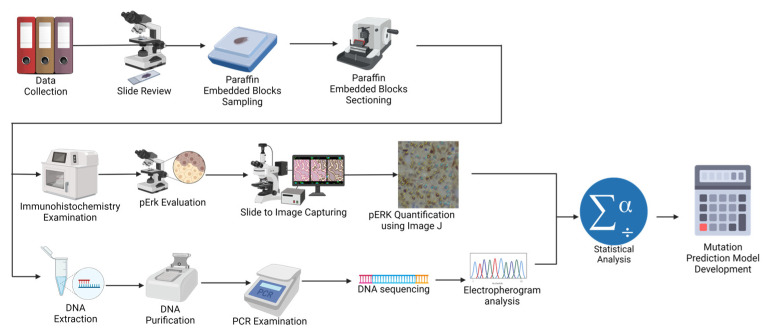
Study protocol. Created with BioRender.com available at https://app.biorender.com/illustrations/63f1d7b1a1db26aa324bb365 (accessed on 8 October 2023).

**Figure 3 biomedicines-11-02803-f003:**
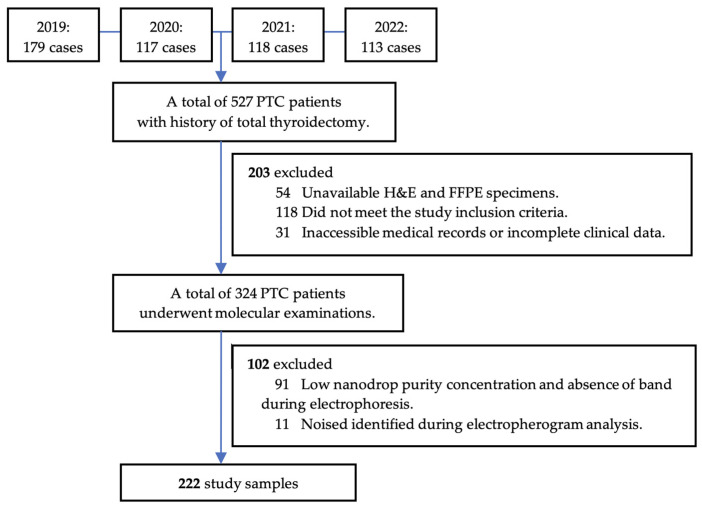
Sample recruitment.

**Figure 4 biomedicines-11-02803-f004:**
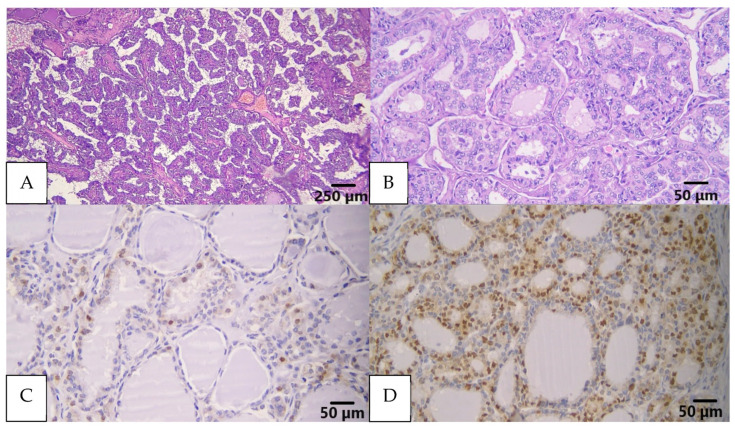
(**A**) Classic subtype of PTC showed papillary structures with fibrovascular core (H&E); (**B**) follicular subtype of PTC, tumor predominantly arranged in a follicular pattern (H&E); (**C**) low pERK1/2 expression, stained brown in the nucleus (IHC pERK1/2); (**D**) high pERK1/2 expression, stained brown in the nucleus (IHC pERK1/2).

**Figure 5 biomedicines-11-02803-f005:**
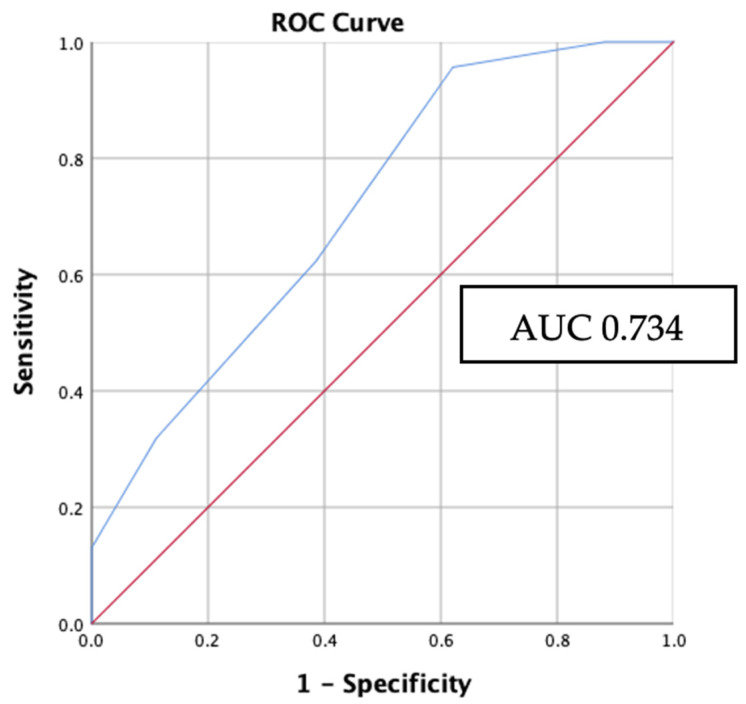
ROC curve of the *BRAF*V600E prediction model. Red line represents the prediction due to chance with AUC 0.5. Blue line represents the model’s performance with AUC 0.734.

**Figure 6 biomedicines-11-02803-f006:**
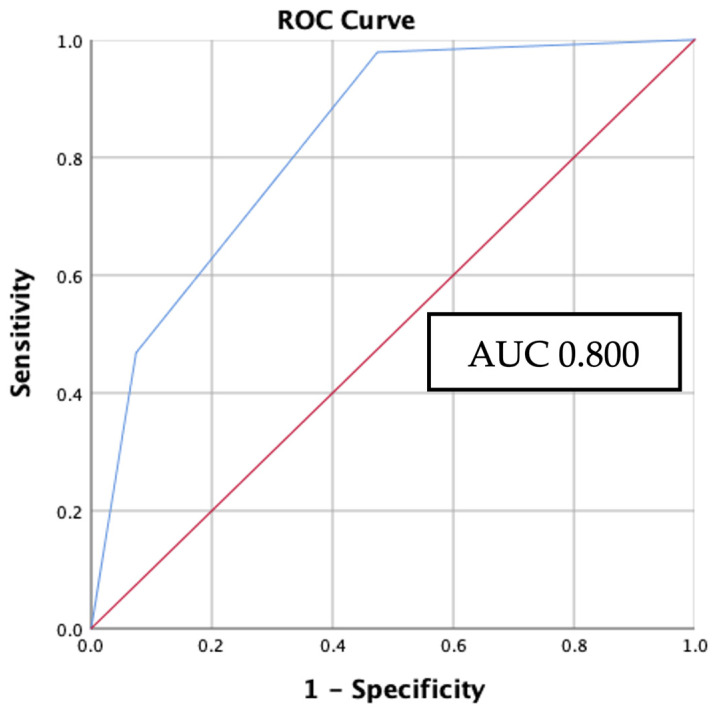
ROC curve of the *RAS* prediction model. Red line represents the prediction due to chance with AUC 0.5. Blue line represents the model’s performance with AUC 0.8.

**Figure 7 biomedicines-11-02803-f007:**
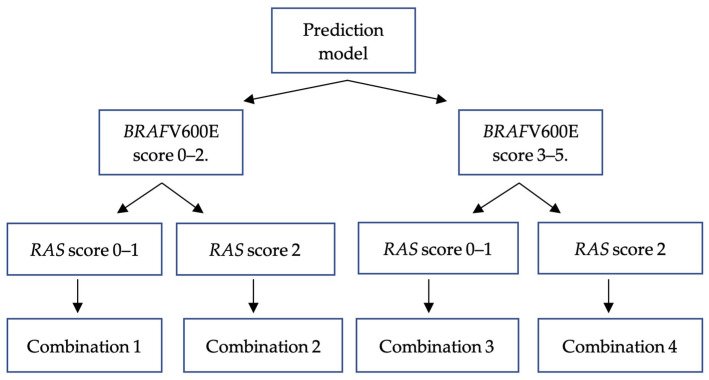
Prediction model of four combination outcomes of *BRAF*V600E and *RAS* mutations.

**Table 1 biomedicines-11-02803-t001:** Correlations between clinico-histopathology characteristics and *BRAF*V600E mutation.

Characteristics	*BRAF*V600E *N* = 64 (%)	Control *N* = 116 (%)	*p*	OR	95% CI
**Clinical features**					
Age (years)					
≥55	17 (35.4)	31 (64.4)	0.981 ^a^	0.992	0.497–1.978
<55	47 (35.6)	85 (64.4)		1.000	Reference
Gender					
Man	18 (36.0)	32 (64.0)	0.938 ^a^	1.027	0.520–2.028
Woman	46 (35.4)	84 (64.6)		1.000	Reference
Clinical stage					
Clinical stage IV	7 (58.3)	5 (41.7)	0.114 ^b^	2.930	0.879–9.766
Clinical stage III	1 (20)	4 (80)		0.523	0.057–4.824
Clinical stage II	13 (43.3)	17 (56.7)		1.601	0.713–3.592
Clinical stage I	43 (32.3)	90 (67.7)		1.000	Reference
Stage group					
Late stage (III–IV)	8 (47.1)	9 (52.9)	0.298 ^a^	1.698	0.621–4.643
Early stage (I–II)	56 (34.4)	107 (65.6)		1.000	Reference
**Histopathology features**					
Tumor size (cm)					
≥4	16 (34.8)	30 (64.2)	0.899 ^a^	0.956	0.474–1.928
<4	48 (35.8)	86 (65.2)		1.000	Reference
Nuclear score					
3	56 (43.4)	73 (56.6)	<0.001 ^a^	4.123	1.796–9.466
2	8 (15.7)	43 (84.3)		1.000	Reference
Capsule					
Absent	52 (44.1)	66 (55.9)	<0.001 ^a^	3.283	1.586–6.794
Present	12 (19.4)	50 (80.6)		1.000	Reference
Histology subtype					
Solid	2 (33.3)	4 (66.7)	<0.001 ^b^	2.938	0.459–18.786
Oncocytic	1 (10)	9 (90)		0.653	0.072–5.878
Classic	21 (37.5)	35 (62.5)		3.525	1.399–8.885
Tall cell	32 (60.4)	21 (39.6)		8.952	3.532–22.690
Follicular	8 (14.5)	47 (85.5)		1.000	Reference
Histology group					
Aggressive	35 (50.7)	34 (49.3)	<0.001 ^a^	2.911	1.544–5.488
Non-aggressive	29 (26.1)	82 (73.9)		1.000	Reference
Multifocality					
Present	50 (35.7)	90 (64.3)	0.934 ^a^	1.032	0.494–2.154
Absent	14 (35.0)	26 (65.0)		1.000	Reference
Lymphovascular invasion					
Present	31 (43.1)	41 (56.9)	0.086 ^a^	1.718	0.924–3.197
Absent	33 (30.6)	75 (69.4)		1.000	Reference
Extrathyroidal extension					
Present	28 (49.1)	29 (50.9)	0.010 ^a^	2.333	1.220–4.463
Absent	36 (29.3)	87 (70.7)		1.000	Reference
Node metastases					
Present	34 (47.2)	38 (52.8)	0.008 ^a^	2.326	1.244–4.349
Absent	30 (27.8)	78 (72.2)		1.000	Reference
pERK1/2 expression					
High (>10%)	25 (51)	24 (49)	0.008 ^a^	2.457	1.253–4.820
Low (<10%)	39 (29.8)	92 (70.2)			Reference

^a^ Chi-squared tests. ^b^ Mann–Whitney U tests.

**Table 2 biomedicines-11-02803-t002:** Correlations between clinico-histopathology characteristics and *RAS* mutation.

Characteristics	*RAS* Mutation *N* = 42 (%)	Control *N* = 116 (%)	*p*	OR	95% CI
**Clinical factors**					
Age (years)					
<55	31 (26.7)	85 (73.3)	0.947 ^a^	1.028	0.461–2.291
≥55	11 (26.2)	31 (73.8)		1.000	Reference
Gender					
Woman	32 (27.6)	84 (72.4)	0.635 ^a^	1.219	0.538–2.764
Man	10 (23.8)	32 (76.2)		1.000	Reference
Clinical stage					
Clinical stage I	33 (26.8)	90 (73.2)	0.981 ^b^	0.458	0.116–1.811
Clinical stage II	5 (22.7)	17 (77.3)		0.368	0.071–1.915
Clinical stage III	0 (0)	4 (100)		0.556	0.310–0.997
Clinical stage IV	4 (44.4)	5 (55.6)		1.000	Reference
Stage group					
Early stage (I–II)	38 (26.2)	107 (73.8)	0.721 ^a^	0.799	0.232–2.746
Late stage (III–IV)	4 (30.8)	9 (69.2)		1.000	Reference
**Histopathology factors**					
Tumor size (cm)					
<4	32 (27.1)	86 (72.9)	0.696 ^a^	1.116	0.490–2.541
≥4	10 (25.0)	30 (75.0)		1.000	Reference
Nuclear score					
2	19 (30.6)	43 (69.4)	0.353 ^a^	1.402	0.686–2.867
3	23 (24.0)	73 (76.0)		1.000	Reference
Capsule					
Present	22 (30.6)	50 (69.4)	0.302 ^a^	1.452	0.715–2.948
Absent	20 (23.3)	66 (76.7)		1.000	Reference
Histology subtype					
Follicular	32 (40.5)	47 (59.5)	<0.001 ^b^	2.648	1.121–6.253
Solid	1 (20)	4 (80)		0.972	0.960–9.799
Oncocytic	0 (0)	9 (100)		1.257	1.082–1.460
Tall cell	0 (0)	21 (100)		1.257	1.082–1.460
Classic	9 (20.5)	35 (79.5)		1.000	Reference
Histology group					
Non-aggressive	41 (33.3)	82 (66.7)	<0.001 ^a^	17.000	2.247–128.615
Aggressive	1 (2.9)	34 (97.1)		1.000	Reference
Multifocality					
Present	28 (23.7)	90 (76.3)	0.163 ^a^	0.578	0.266–1.255
Absent	14 (35.0)	26 (65.0)		1.000	Reference
Lymphovascular invasion					
Present	15 (25.4)	41 (73.2)	0.966 ^a^	1.016	0.486–2.124
Absent	27 (26.5)	75 (73.5)		1.000	Reference
Extrathyroidal extension					
Absent	35 (28.5)	87 (71.3)	0.270 ^a^	1.667	0.668–4.157
Present	7 (19.4)	29 (80.6)		1.000	Reference
Node metastasis					
Absent	26 (25.0)	78 (75.0)	0.532 ^a^	0.792	0.380–1.649
Present	16 (29.6)	38 (70.4)		1.00	Reference
pERK1/2 expression					
High (>10%)	28 (53.8)	24 (46.2)	<0.001 ^a^	7.667	3.503–16.778
Low (<10%)	14 (13.2)	92 (86.8)			Reference

^a^ Chi-squared tests. ^b^ Mann–Whitney U tests.

**Table 3 biomedicines-11-02803-t003:** Logistic regression of the *BRAF*V600E prediction model.

Variables	B Coefficient	SE	Wald	*p*	adjOR	95% CI	B/SE	Score
Nuclear score (3)	1.213	0.480	6.375	0.012	3.364	1.312–8.626	2.527	1
Capsule (absent)	0.975	0.412	5.605	0.018	2.651	1.183–5.941	2.366	1
Histology subtypes (aggressive)	0.858	0.375	5.218	0.022	2.358	1.130–4.921	2.288	1
pERK1/2 (>10%)	1.460	0.410	12.668	≤0.001	4.308	1.927–9.627	3.560	2

**Table 4 biomedicines-11-02803-t004:** Probability, sensitivity, and specificity of the outcomes of the *BRAF*V600E prediction model.

Total Score	Probability	Sensitivity	Specificity
0	5%	100%	0%
1	12.33%	100%	12%
2	25.25%	95%	39%
3	43%	63%	65%
4	62%	30%	94%
5	82%	14%	100%

**Table 5 biomedicines-11-02803-t005:** Logistic regression of the *RAS* mutation prediction model.

Variables	B Coefficient	SE	Wald	*p*	adjOR	95% CI	B/SE	Score
pERK1/2 (>10%)	2.101	0.430	23.865	≤0.001	8.171	3.518–18.981	4.886	1
Histology subtype (follicular)	1.628	0.454	12.877	≤0.001	5.092	2.092–12.387	3.585	1

**Table 6 biomedicines-11-02803-t006:** Probability, sensitivity, and specificity of the outcomes of the *RAS* prediction model.

Total Score	Probability	Sensitivity	Specificity
0	5%	100%	0%
1	27%	98%	48%
2	70%	45%	91%

**Table 7 biomedicines-11-02803-t007:** Multinomial analysis between the four combination outcomes and mutational status.

Model Outcome	*BRAF*V600E *N* (%)	*p*	adjOR	95% CI	*RAS* *N* (%)	*p*	adjOR	95% CI	Control *N* (%)
Comb. 1	22 (34.4)	Ref	Ref	Reference	14 (33.3)	Ref	1.00	Reference	68 (58.6)
Comb. 2	2 (3.1)	0.882	0.883	0.171–4.568	7 (16.7)	0.010	4.857	1.470–16.049	7 (6)
Comb. 3	37 (57.8)	≤0.001	3.091	1.594–5.995	9 (21.4)	0.725	1.181	0.467–2.989	37 (31.9)
Comb. 4	3 (4.7)	0.295	2.318	0.481–11.168	12 (28.6)	≤0.001	14.571	4.095–51.855	4 (3.4)

Comb. = combination.

**Table 8 biomedicines-11-02803-t008:** Correlations between combination groups with clinical stage and node metastasis.

Model Outcome	Clinical Stage					Node Metastasis				
	Early *N* (%)	Late *N* (%)	*p*	adjOR	95% CI	No *N* (%)	Yes *N* (%)	*p*	adjOR	95% CI
Comb. 1	99 (95.2)	5 (4.8)	0.008 ^a^	1.000	Reference	76 (73)	28 (27)	<0.001 ^a^	1.000	Reference
Comb. 2	16 (100)	0 (0)		1.162	1.079–1.250	14 (87.5)	2 (12.5)		0.388	0.083–1.815
Comb. 3	67 (80.7)	16 (19.3)		4.728	1.653–13.525	32 (38.5)	51 (61.5)		4.326	2.330–8.033
Comb. 4	17 (89.5)	2 (10.5)		2.329	0.418–12.991	12 (63.1)	7 (36.9)		1.583	0.566–4.426

Comb. = combination. ^a^ Mann–Whitney U tests.

## Data Availability

The dataset utilized in this study is accessible upon request from the author responsible for correspondence.

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
