# Peer review of "Developing Models to Predict BRAFV600E and RAS Mutational Status in Papillary Thyroid Carcinoma Using Clinicopathological Features and pERK1/2 Immunohistochemistry Expression"

_biomedicines, 2023, doi:10.3390/biomedicines11102803_

Round 1

Reviewer 1 Report

This research analyzed the histopathological and mutational status of BRAF and RAS in papillary thyroid carcinoma following the TCGA/WHO2022 classification. The authors analyzed a large series of cases. The mutational status and the histochemistry of pERK1/2 were correlated with several variables. As a result, the mutational status was of each of the two main subtypes were predicted. This part of the manuscript is easy to understand, and validates previous knowledge/findings.

Later, the authors make a more elavorated prediction score with "combination". This part is more difficult to translate into the clinical practice. Do the combination strategy provides an advantage to the previous analysis?

Additional comments:

(1) Line 55. Could you please confirm the prevalence rate of 25-82%. This percentage has a broad range.

(2) Line 56. Regarding mutually exclusive. Do you mean that there are no cases with double mutation BRAF V600E and RAS? What about the non-BRAF/non-RAS-like?

(3) Lines 55-68. Could you please add a figure with the pathway, showing the BRAF and RAS molecules, the connections, the effect of the tumation, etc.? This figure could help the reader understand the pathological mechanism better.

(4) This manuscript makes a revision and pdate from the 2022 WHO classification:

Jung CK, Bychkov A, Kakudo K. Update from the 2022 World Health Organization Classification of Thyroid Tumors: A Standardized Diagnostic Approach. Endocrinol Metab (Seoul). 2022 Oct;37(5):703-718. doi: 10.3803/EnM.2022.1553. Epub 2022 Oct 4. PMID: 36193717; PMCID: PMC9633223.

Figure 1 is very informative.

The BRAFV600E-like molecular profile includes the BRAFV600E mutation and gene fusions involving BRAF, RET, and neurotrophic receptor tyrosine kinase 1/3 (NTRK1/3).

RAS-like molecular profiles include NRAS, HRAS, KRAS, EIF1AX, enhancer of zeste 1 polycomb repressive complex 2 subunit (EZH1), Dicer 1, ribonuclease III (DICER1), phosphatase and tensin homolog (PTEN) mutations, BRAFK601E, and gene fusions involving peroxisome proliferator-activated receptor gamma (PPARG) and THADA.

When the three-group molecular classification is applied, PAX8::PPARG gene fusion and mutations of EIF1AX, EZH1, IDH1, SOS1, SPOP, DICER1, and PTEN genes are classified as a non-BRAFV600E-/non-RAS-like group.

Question: Is the TCGA classification comparable/equivalent to the one of WHO2022?

(5) Line 100. Could you please provide a descrition of the PTC-nuclear score? You may add this score in the Appendix.

(6) What are the positive and negative controls of teh pERK1/2 immunostainings?

(7) What is the catalogue number of the normal horse serum?

(8) Lines 129-133. Do the Sanger sequencing cover the mutations of the TCGA/WHO2022?

(9) Lines 144 - 145. Could you please explain with more detail how the scoring system was made?

(10) As I understand, in both BRAFV600E and RAS mutated cases the immunohistochemistry of pERK1/2 was high, but in the unmutated cases the staining was lower. Is it possible to show histological images?

(11) In Table 3. Why pERK1/2 is given a score of 2, but the other 3 variables have a score of 1? Is it related to the adjusted Odds Ratio?

(12) Regarding Table 4. As I understand, the predictive factors are additive, the higher the score, the higher the probability of being BRAFV600E mutated?

(13) Section 3.5 makes a step further into the probabilistic method. What is the usefulness of this combination prediction model?

Author Response

Dear Reviewer 1,

We would like to say our sincere gratitude for taking the time to review our submitted manuscript.

Please see the attachment below for the detailed response regarding your review.

Thank you so much for the invaluable suggestion.

Best regards,

Authors

Reviewer 2 Report

Specific comments to the authors

The authors Harahap et al. of the submitted manuscript „Developing models to predict BRAFV600E and RAS mutational status in papillary thyroid carcinoma.” studied the possibility to predict the BRAFV600E and RAS mutational status in papillary thyroid carcinoma (PTC) based on clinicopathological characteristics and pERK1/2 expression variations between BRAF-like and RAS- like PTCs.

Based on their clinico-pathological investigations the authors could detect independent predictors for the BRAFV600E mutation modell (nuclear score of 3, the absence of capsules, an aggressive histology variant, and pERK1/2 levels exceeding 10%) as well as for the RAS mutation predictive modell (follicular histology variant and pERK1/2 expression >10%), which could be significantly combined, too. The authors of the manuscript postulated that different prediction models indicate variations in biological behaviour between BRAF-like and RAS-like PTCs.

Overall, the manuscript gives some interesting aspects to predict BRAFV600E and RAS mutational status in PTCs based on clinico-pathological characteristics. The manuscript (including presentation) is mostly comprehensible and convincing. The methods are well described. Although the results and discussion are clear presented, some minor changes could be performed by the authors (see specific comments) to improve the manuscript.

In conclusion, the presented data are interesting. After incorporating the mentioned specific comments (see below) the manuscript has the potency to be accepted.

Specific comments

Title: Please add the basis for the prediction modell (clinico-pathological characteristics) in the title.

Abstract: Please clarify the kind of presented clinical investigation (retrospective versus prospective, randomized etc.) in the abstract. The conclusion “The different prediction models indicate variations in biological behaviour between BRAF-like and RAS-like PTCs, necessitating adjustments in treatment approaches” is largely speculative, since no clinical endpoints of this patient group is investigated by the authors.

Material and Methods: Please include the review and resolution of the local ethics committee.

Results:

# Table 1 and 2: Regarding the variable “Histology group” please define the values “Aggressive” and “Non-aggressive” in more detail in relation to the sentence “We excluded cases with aggressive features such as a high mitosis index and necrosis”.

Table 7: The findings must be correlated to clinical endpoints to validate the prognostic value of the findings, too.

Discussion: Regarding the sentence “We identified three features that emerged as significant predictors of the presence of the BRAFV600E mutation” the authors give four features: (i) nuclear score of 3, (ii) the aggressive histology variants, (iii) the lack of tumor capsule, and (iv) an expression level of pERK1/2 greater than 10%”. Please correct adequately. Regarding the sentence “The current research demonstrates a positive correlation between higher total scores and an increased probability of observing a BRAFV600E mutation in PTC” please add corresponding correlation analysis in the results chapter, too. Please define the term “low-grade tumors” in more detail. The sentence “Hitherto, this study is the sole research that has constructed a predictive model pertaining to gene mutations in PTC, owing to the routine implementation of molecular examination in developed countries.” is the key message of this study and should be more emphasized for such countries and laboratories which are not able to carry out such additional molecular anaylsis of PTCs.

 Minor editing of English language required.

Author Response

Dear Reviewer 2,

We would like to say our sincere gratitude for taking the time to review our submitted manuscript.

Please see the attachment below for the detailed response regarding your review.

Thank you so much for the invaluable suggestion.

Best regards,

Authors

Reviewer 3 Report

The manuscript "Developing models to predict BRAFV600E and RAS mutational status in papillary thyroid carcinoma" describes an interesting research evaluating the mutations in thyroid carcinoma.

I think that this manuscript is in line with journal guidelines and some aspects are very relevant for research in this field. Some minor points need to be revised:

a clear figure describing the steps of the protocol must be useful for readeness.

statistical analysis is missing.

chemometric analysis could be useful for interpretation of the results.

Author Response

Dear Reviewer 3,

We would like to say our sincere gratitude for taking the time to review our submitted manuscript.

Please see the attachment below for the detailed response regarding your review.

Thank you so much for the invaluable suggestion.

Best regards,

Authors
